# Programmed cell death-1 is involved with peripheral blood immune cell profiles in patients with hepatitis C virus antiviral therapy

Miyabi Miura[1☯], Michiko Nishino[1☯], Kazunori Kawaguchi[1]*, Shihui Li[1], Tetsuro Shimakami[1], Toshikatsu Tamai[1], Hidetoshi Nakagawa[1], Takeshi Terashima[1], Noriho Iida[1], Hajime Takatori[1], Kuniaki Arai[1], Yoshio Sakai[1], Tatsuya Yamashita[1], Masao Honda[1], Shuichi Kaneko[2], Eishiro Mizukoshi[1], Taro Yamashita[1]

1 Department of Gastroenterology, Kanazawa University Graduate School of Medical Sciences, Kanazawa, Ishikawa, Japan, 2 Department of Information-Based Medicine Development, Kanazawa University Graduate School of Medicine, Ishikawa, Japan

☯ These authors contributed equally to this work.
* kawaguchi@m-kanazawa.jp

**Data Availability Statement:** All relevant data are within the manuscript and its Supporting Information files.

## Abstract

Mutations in the non-structural protein regions of hepatitis C virus (HCV) are a cause of a non-sustained virological response (SVR) to treatment with direct-acting antivirals (DAAs) for chronic hepatitis; however, there are non-SVR cases without these mutations. In this study, we examined immune cell profiles in peripheral blood before and after ombitasvir/paritaprevir/ritonavir treatment and screened for genes that could be used to predict the therapeutic effects of DAAs. Fluorescence-activated cell sorting analysis indicated that the median frequencies of programmed cell death-1-positive (PD-1+) effector regulatory T cells (eTregs), PD-1+CD8+ T cells, and PD-1+Helper T cells were decreased significantly in SVR cases, but without significant changes in non-SVR cases. The frequency of PD-1+ naïve Tregs was significantly higher in the SVR group than in the non-SVR group before and after treatment. Similar results were found in patients treated with other DAAs (e.g., daclatasvir plus asunaprevir) and supported an immune response after HCV therapy. RNA-sequencing analysis indicated a significant increase in the expression of genes associated with the immune response in the SVR group, while genes related to intracellular and extracellular signal transduction were highly expressed in the non-SVR group. Therefore, we searched for genes associated with PD-1+ eTregs and CD8+ T cells that were significantly different between the SVR and non-SVR groups and found that T-box transcription factor 21 was associated with the non-SVR state. These results indicate that PD-1-related signaling pathways are associated with a non-SVR mechanism after DAAs treatment separate from mutation-related drug resistance.

**Funding:** The research leading to these results received funding from AbbVie Inc. The funders had no role in study design, data collection and analysis, decision to publish, or preparation of the manuscript

**Competing interests:** The authors have declared that no competing interests exist.

## Introduction

Hepatitis C virus (HCV) induces chronic liver disease, which can lead to cirrhosis and hepato-cellular carcinoma (HCC). It is estimated that 71 million people worldwide are chronically infected with HCV, and approximately 400,000 people die each year from HCV-related liver disease [1]. Interferon (IFN) alone or IFN plus ribavirin has traditionally been used as antiviral therapy for HCV; however, the rate of a sustained virological response (SVR) is low with this approach, especially in genotype 1b patients. The IL28B gene polymorphism is a predictor of the effect of IFN on the SVR [2,3], and this genetic biomarker is used clinically for any IFN-based therapies. The development of direct-acting antivirals (DAAs) has enabled the majority of patients to achieve an SVR, including protease inhibitors targeting the non-structural 3 (NS3) region, NS5A inhibitors, and NS5B polymerase inhibitors [4–6]. DAAs and their clinical application have been advanced continually and their effectiveness has been established with respect to genotype and major reported mutations; moreover, the use of DAAs has overcome the problems associated with the IL28B polymorphism.

Patients with an SVR after antiviral therapy have reduced liver inflammation or liver fibrosis, resulting in a lower incidence of mortal events such as liver cirrhosis-related complications and HCC [7]. Nevertheless, the incidence of HCC is still high in patients with an SVR. One of the reasons for this is the residual fibrosis and liver damage caused by other factors such as lipotoxicity under fatty liver disease such as non-alcoholic fatty liver disease or non-alcoholic steatohepatitis and other risk factors such as alcohol consumption, advanced age, male sex, and past infection with hepatitis B virus [8].

In addition, liver inflammation as a result of HCV infection and residual inflammation after antiviral therapy are induced by a specific immune regulatory mechanism, which might lead to the development of HCC [9]. Therefore, we consider that immune changes may have a significant impact on carcinogenesis, even after achieving an SVR. It is also important to examine the immune-regulatory effect of DAAs on this mechanism in detail because it is linked to the risk of HCC in patients with an SVR or non-SVR [10,11]. Despite basic experiments and the clinical administration of various DAAs, neither the immune response nor the correlation linked to the efficacy of DAAs has been evaluated fully, except for the known mutations such as in the NS5A region. Although the L31 and Y93 mutations in the NS5A region are associated with a non-SVR, other viral and host genetic factors involved in this process remain unknown [12].

Programmed cell death-1 (PD-1) is associated with cancer immune checkpoint mechanisms, and anti-PD-1 antibodies have been used widely for immunotherapy in various types of cancer, including HCC [13,14]. For example, combination therapy with the anti-PD-L1 antibody atezolizumab plus the anti-vascular endothelial growth factor bevacizumab is effective for HCC patients [15]. Moreover, these immune checkpoint factors repress the antiviral immune reaction induced by PD-1 or its ligand PD-L1. For example, PD-L1 is overexpressed in Epstein–Barr virus infection and related carcinogenesis [16,17]. Furthermore, some cancer-related viruses are associated with PD-L1 expression such as lymphoma [18]. However, there are few reports examining viral infection and the outcome of antiviral therapy with PD-1 dynamics. Some reports have found an association in patients with an SVR under HCV infection and PD-1 dynamics in lymphoid cells [19,20]. However, there is little information regarding the differences between SVR and non-SVR patients.

For this reason, we considered it important to examine samples from HCV antiviral therapy patients, especially immune cells. Since a wide range of DAAs targeting HCV have been introduced, we conducted a detailed analysis of the factors that prevented an SVR by comparing SVR and non-SVR cases using peripheral blood mononuclear cells (PBMCs). Using

multicenter data and PBMCs from patients who underwent ombitasvir/paritaprevir/ritonavir (OBV/PTV/r) therapy, the factors associated with a non-SVR with DAAs therapy in the absence of well-known mutations were clarified using gene expression analysis [21,22].

In this study, we investigated the factors predicting the efficacy of DAAs treatment by searching for the presence or absence of gene mutations associated with a non-SVR and changes in the peripheral blood immune cell profiles of patients before and after DAAs therapy. We analyzed patients who took OBV/PTV/r therapy and clarified their immune profiles and gene expression patterns using PBMCs. Moreover, we also analyzed the immune profiles of patients who were administered the DAAs daclatasvir (DCV) and asunaprevir (ASV). We focused on the differences between SVR and non-SVR patients, in addition to known SVR status-associated mutations.

## Materials and methods

### Patients and DAAs therapy

Two hundred and seven patients treated with OBV/PTV/r in our region from March 1, 2016 to March 31, 2018 participated in this study; 194 patients achieved an SVR and 13 patients did not achieve an SVR, with an SVR rate of approximately 90%. Following the exclusion of patients who did not complete therapy due to side effects and those with a lack of RNA and PBMC samples or data, we included 27 SVR cases and four non-SVR cases (11 male and 20 female; mean age, 65.2 years) for analysis (S1 Fig). Among the 31 OBV/PTV/r-treated cases, 17 had not received previous HCV antiviral treatment, six had received pegylated IFN/ribavirin treatment, seven had received IFN monotherapy, and one case was other. In the histological analysis of 16 cases, 4 had liver cirrhosis with stage 4 fibrosis. In the cases without histological analysis, there were 12 with chronic hepatitis and 3 with liver cirrhosis according to a non-invasive fibrosis assessment method (Table 1). Three SVR cases had a history of HCC, which had already been treated, with no recurrence. There were no significant changes in serum aspartate aminotransferase, alanine aminotransferase, total bilirubin, albumin, and platelet counts among the SVR and non-SVR cases before OBV/PTV/r treatment; however, these data were improved in the SVR cases after OBV/PTV/r treatment. Among the cases that received OBV/PTV/r treatment, we did not observe resistance mutations such as Y93 and L31 in the NS5A position. We assessed the degree of chronic hepatitis by various methods. We determined the degree of liver fibrosis by performing transient elastography, while the FIB-4 index calculation was used in the cases without histological assessment; we defined liver cirrhosis as greater than 17 kPa for transient elastography, and greater than 2.67 for the FIB-4 index calculation. The cases received oral doses of 25 mg OBV, 150 mg PTV, and 100 mg ritonavir once a day for 12 weeks.

We also analyzed 462 DCV- and ASV-treated patients from October 1, 2014, to April 30, 2017, 413 with an SVR and 49 without an SVR, and we could analyze the immune profiles of PBMCs from 30 SVR cases and 10 non-SVR cases (S2 Fig). These cases were administered an oral dose of 60 mg DCV once a day and 100 mg ASV twice a day for 24 weeks. FIve of these cases had a tyrosine to histidine mutation at amino acid 93 (Y93H) and one case had a lysine to methionine mutation at amino acid 31 (L31M) in the HCV NS5A region, whereas the Y93H/L31M mutation was not observed in the OBV/PTV/r-treated cases (S1 Table).

Written informed consent was obtained from all patients before the start of DAAs therapy. We performed essential clinical blood tests and image analysis, such as liver enzymes, HCV markers, and genotypes as well as PBMC isolation. We assessed HCV-RNA disappearance to determine SVR or non-SVR status at 12 or 24 weeks after the end of the DAAs administration period.

**Table 1. Patient characteristics for OBV/PTV/r treatment.**

|  |  |  | | All cases (*n* = 31) |
| --- | --- | --- | --- | --- |
| Age (years) |  |  | | 65.2 (42–82) |
| Gender (male/female) |  |  | | 11/20 |
| Pre-treatment: naïve/IFN+RBV/IFN monotherapy/other |  |  | | 17/6/7/1 |
| Chronic hepatitis/cirrhosis |  |  | | 24/7 |
| Histologically analyzed chronic hepatitis/cirrhosis |  |  | | 16/4 |
| Non-histologically analyzed chronic hepatitis/cirrhosis |  |  | | 12/3 |
| History of HCC: yes/no |  |  | | 3/28 |
| SVR/non SVR |  |  | | 27/4 |

|  | SVR (*n* = 27) | non SVR (*n* = 4) | *P*-value |
| --- | --- | --- | --- |
| Age (years) | 64.8 (42–82) | 67.8 (54–77) | NS |
| Gender (male/female) | 10/17 | 1/3 | NS |
| Pre-treatment: Naïve/IFN+RBV/IFN/other |  |  |  |
| Chronic hepatitis/cirrhosis | 16/6/5/0 | 1/0/2/1 | NS |
| Liver biopsy: yes/no | 21/6 | 3/1 | NS |
| History of HCC: yes/no | 15/12 | 1/3 | NS |
| AST (IU/L) | 3/24 | 0/4 | NS |
| ALT (IU/L) | 46.9 ± 6.4 | 26.3 ± 5.0 | NS |
| Total bilirubin (mg/dL) | 42.3 ± 5.0 | 27.5 ± 7.7 | NS |
| Albumin (g/dL) | 0.9 ± 0.1 | 0.8 ± 0.1 | NS |
| Platelet count (×$10^4$/μL) | 3.9 ± 0.1 | 4.5 ± 0.3 | NS |
| Post-treatment: | 15.5 ± 1.6 | 15.7 ± 2.4 | NS |
| AST (IU/L) | 23.6 ± 2.1 | 37.3 ± 8.2 | <0.05 |
| ALT (IU/L) | 16.0 ± 1.4 | 48.3 ± 18.2 | NS |
| Total bilirubin (mg/dL) | 0.8 ± 0.1 | 0.8 ± 0.2 | NS |
| Albumin (g/dL) | 4.2 ± 0.1 | 4.6 ± 0.1 | <0.01 |
| Platelet count (×$10^4$/μL) | 22.6 ± 5.2 | 15.7 ± 3.2 | NS |

Abbreviations: ALT, alanine aminotransferase; AST, aspartate aminotransferase; HCC, hepatocellular carcinoma; IFN, interferon; NS, not significant; OBV/PTV/r, ombitasvir/paritaprevir/ritonavir; RBV, ribavirin; SVR, sustained virological response.

This study was conducted in accordance with the Declaration of Helsinki and was approved by the ethics committee at each participating hospital (approval no. 1639, 2038). Medical records from March 1, 2016 to July 31, 2023 were assessed for research purposes.

## PBMC samples

PBMCs were isolated according to a previously established method from 29 mL peripheral blood drawn from the patients before the start of DAAs therapy and at 12 weeks after finishing treatment. The PBMCs were suspended in CELLBANKER 1 (ZENOAQ, Fukushima, Japan) containing 80% fetal calf serum and 10% dimethyl sulfoxide and stored at −150°C in a liquid nitrogen tank before analysis.

## Analysis of peripheral immune cells

To determine the immune cell profiles, multi-color fluorescence-activated cell sorting (FACS) analysis was performed using antibodies targeting the following proteins: CD3, CD4, CD8, 4-1BB, and OX40; T cell marker, CD14 and CD15; granulocyte marker, CD25; IL-2 receptor, CD80 and CD45RA; B cell marker, HLA-DR; MHC-class II marker, FoxP3; regulatory T cells (Tregs) marker, CTLA-4, PD-1, and PD-L1; immune checkpoint marker, CCR4, CCR6, and CXCR3; and chemokine receptor marker (Becton Dickinson, Franklin Lakes, NJ). Flow cytometry was performed using a FACS Aria II (Becton Dickinson). We investigated immune

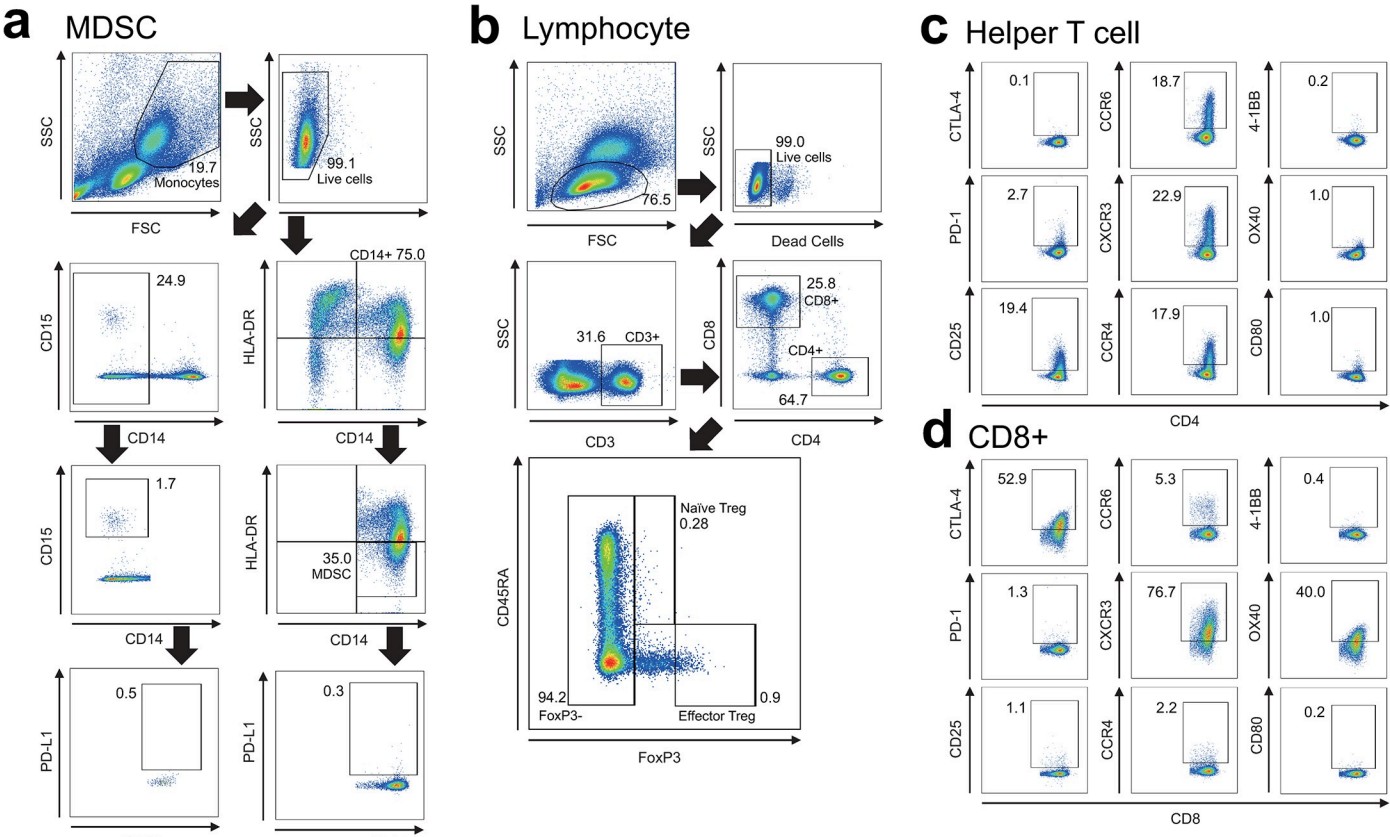

**Fig 1. Peripheral blood immune cell profiles.** (a) The frequencies of MDSCs were measured by multicolor FACS analysis using antibodies targeting the following proteins: CD14, CD15, HLA-DR and PD-L1. (b) The frequencies of CD4+ T cells, CD8+ T cells, and effector Tregs were measured by multicolor FACS analysis using antibodies targeting the following proteins: CD3, CD4, CD8, CD45RA, and FoxP3. Effector Tregs were defined as CD3+, CD4+, CD45RA− and FoxP3-high cells; naïve Tregs were defined as CD3+, CD4+, CD45RA+, and FoxP3-low cells; and non-Tregs were defined as CD3+, CD4+, CD45RA+/−, and FoxP3− cells. (c) Expression levels of CD25, CTLA-4, PD-1, CCR4, CXCR3, CCR6, CD80, OX40, and 4-1BB were measured in non-Tregs. (d) Expression levels of CD25, CTLA-4, PD-1, CCR4, CXCR3, CCR6, CD80, OX40, and 4-1BB were measured in CD8+ T cells. FSC, forward scatter; SSC, side scatter.

cell profiles in peripheral blood before and after OBV/PTV/r treatment. In this study, we divided myeloid-derived suppressor cells (MDSCs) in peripheral blood into CD14+CD15-MDSCs and CD14-CD15+ MDSCs according to CD14, CD15, and HLA-DR expression levels, and measured the frequency of each type. We also measured PD-L1 expression levels in each fraction (Fig 1A). For T cells, we isolated CD8+ T cells and CD4+ T cells, and from the CD4+ T cells, we further isolated CD45RA- effector Tregs that strongly expressed FoxP3, CD45RA+ naïve Tregs that weakly expressed FoxP3, and CD4+ T cells that were negative for FoxP3 (defined as helper T cells) for investigation (Fig 1B). We also measured the expression levels of CTLA-4, PD-1, CD25, CCR6, CXCR3, CCR4, 4-1BB, OX40, and CD80 in the different T cell fractions (Fig 1C and 1D). Using these FACS analyses, we investigated whether the immune cell profiles were changed after DAAs treatment.

## RNA-sequencing (seq) analysis and quantitative PCR (qPCR)

We isolated total RNA from PBMCs collected in PAXgene (Becton Dickinson) using a PAX-gene Blood RNA Kit (Qiagen, Valencia, CA). An estimated $1.2 \times 10^7$ to $2.8 \times 10^7$ cells per sample were collected during blood sampling. We evaluated the quality of the isolated RNA, and performed cDNA synthesis. We prepared and sequenced libraries using a next-generation

sequencer for bioinformatics analysis (RIKEN GENESIS, Tokyo, Japan). Using these isolated samples, we performed qPCR to confirm specific gene expression.

## Statistical and gene pathway analyses

Data are expressed as the median ± standard error (SE) for immune cell analysis. We used Fisher's exact test (two-sided *P*-value) and the unpaired Student's *t*-test to analyze the clinical factors of the patients. Using the RNA-seq data, we calculated gene expression before and after DAAs treatment and compared the SVR and non-SVR cases. By classifying these groups, we performed pathway analysis using MetaCore online software (Clarivate, Philadelphia, PA).

## Results

### PD-1 levels are decreased after DAA therapy

We found significant changes in the profiles of some peripheral blood immune cells after OBV/PTV/r treatment compared with before treatment (Fig 2); however, there was no change in the profiles of other immune cells (S3 Fig). The median frequencies of the following cells decreased significantly after OBV/PTV/r treatment compared with before treatment: effector

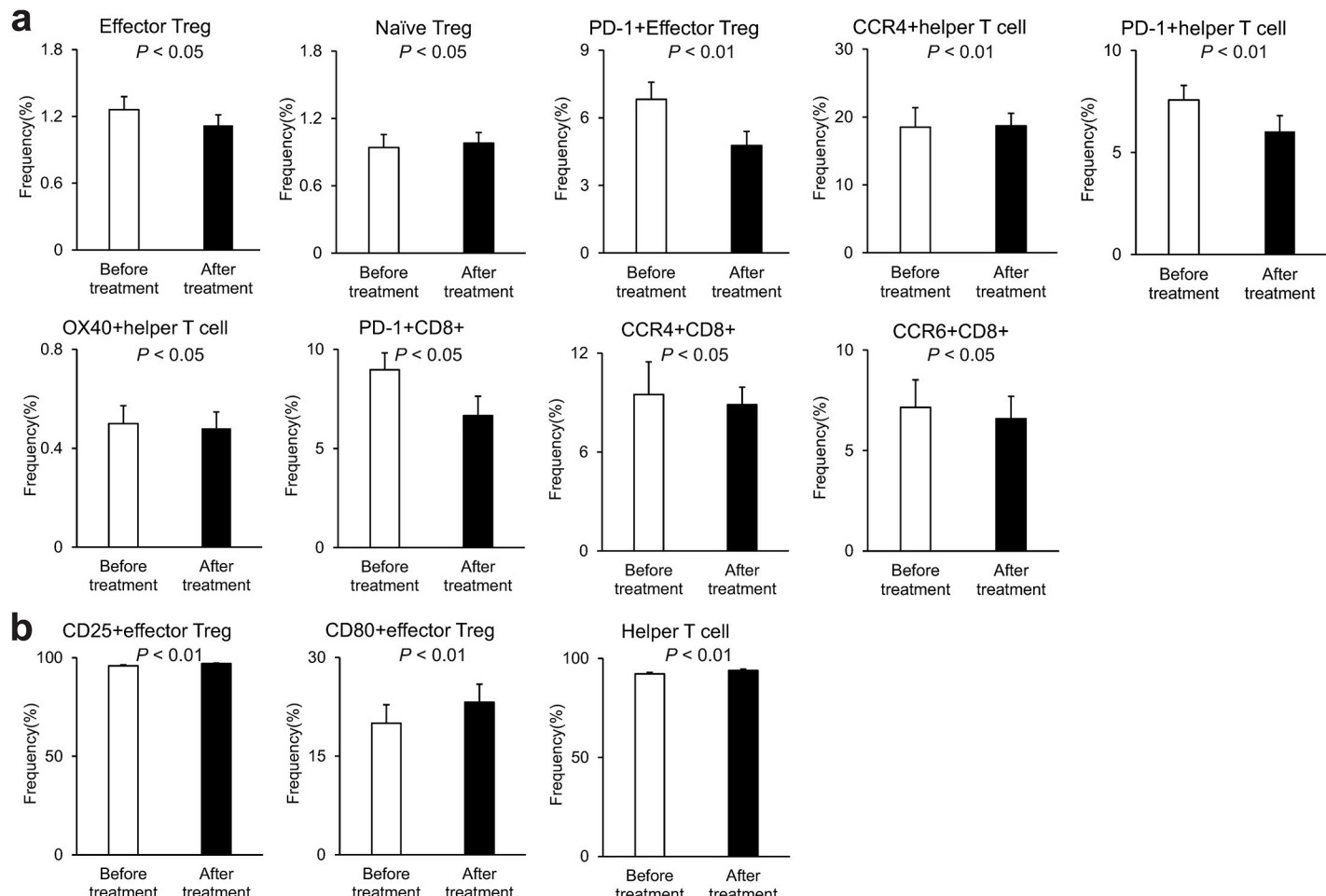

**Fig 2. Frequencies of peripheral blood immune cells with each profile in all patients before and after OBV/PTV/r treatment.** Only significantly changed profiles are listed. (a) Significantly decreased profiles after treatment. (b) Significantly increased profiles after treatment. Scale bars are the median ± SE.

Tregs ($P < 0.05$), naïve Tregs ($P < 0.01$), helper T cells ($P < 0.01$), PD-1+ effector Tregs ($P < 0.01$), PD-1+ helper T cells ($P < 0.01$), CCR4+ helper T cells ($P < 0.01$), OX40+ helper T cells ($P < 0.05$), PD-1+CD8+ T cells ($P < 0.01$), CCR4+CD8+ T cells ($P < 0.05$), and CCR6+CD8+ T cells ($P < 0.05$) (Fig 2A). In contrast, the median frequencies of the following cells increased significantly after treatment compared with before treatment: CD25 + effector Tregs ($P < 0.01$), CD80+ effector Tregs ($P < 0.01$), and helper T cells ($P < 0.01$) (Fig 2B). Most of these patterns were significantly different in the paired comparisons for each patient (S4a and S4B Fig). Other profiles, such as those including PD-L1, were not changed significantly before and after treatment (S3A and S3B Fig). We investigated the changes of the immune cell profiles after OBV/PTV/r between the SVR and non-SVR groups. In the SVR group, the median frequencies of the following cells decreased significantly after treatment compared with before treatment: effector Tregs ($P < 0.05$), PD-1 + effector Tregs ($P < 0.01$), PD-1+ helper T cells ($P < 0.01$), CCR4+ helper T cells ($P < 0.01$), OX40+ helper T cells ($P < 0.05$), PD-1+CD8+ T cells ($P < 0.05$), CCR4+CD8 + T cells ($P < 0.05$), and CCR6+CD8+ T cells ($P < 0.05$) (Fig 3A). In contrast, the median frequencies of the following cells increased significantly after treatment compared with before treatment: CD25+ effector Tregs ($P < 0.01$), CD80+ effector Tregs ($P < 0.01$), and helper T cells ($P < 0.01$) (Fig 3B). Most of these patterns were significantly different in the paired comparison analysis for each patient (S5A and S5B Fig). However, there was no significant change in the immune profiles of the non-SVR cases before and after treatment (S6 Fig). We also observed a similar tendency for each marker in the DCV/ASV-treated cases, but some were not statistically significant (S7 Fig). PD-1+ effector Tregs were significantly decreased in all cases (S7A Fig) and SVR cases (S7B Fig), and other PD-1-related markers were decreased but without statistical significance. There was no change in the immune profiles of non-SVR cases treated with DCV/ASV, including PD-1 markers (S7C Fig). From these results, we found a decrease in the immune profiles of PD-1+ effector Tregs, PD-1+ helper T cells, and PD-1+CD8+ T cells after successful HCV antiviral treatment.

## PD-1 expression differs before DAA therapy between SVR and non-SVR cases

In the SVR group before OBV/PTV/r treatment, we found that the frequencies of CD14-CD15 + MDSCs and PD-1+ naïve Tregs were significantly increased compared with the non-SVR group (Fig 4A). On the other hand, after OBV/PTV/r treatment, the frequencies of CD25 + effector Tregs, PD-1+ naïve Tregs, CXCR3+ naïve Tregs, OX40+ naïve Tregs, and CXCR3 + helper T cells were significantly increased in the SVR group compared with the non-SVR group (Fig 4B). In contrast, in the non-SVR group both before and after OBV/PTV/r treatment, the frequencies of CTLA4+CD8+ T cells and OX40+CD8+ T cells were significantly increased compared with the SVR group. For DCV/ASV treatment, there was a different pattern before treatment in the profiles of CTLA4+CD8+ T cells and OX40+CD8+ T cells between SVR and non-SVR cases with the Y93H/L31M mutation (S8A Fig) and without the Y93H/L31M mutation (S8B Fig) compared with OBV/PTV/r treatment. However, these profiles were similar regardless of the presence or absence of the Y93H/L31M mutation, in that the PD-1+ naïve Treg and PD-1+ effector Treg profiles were similar, but not statistically significant, including after treatment with the Y93H/L31M mutation (S8C Fig) and without the Y93H/L31M mutation (S8D Fig). From these comparisons, we found that there were some differences in the PD-1-related immune cell profile between SVR and non-SVR cases, indicating that PD-1 expression in Tregs contributes to HCV eradication by antiviral therapy.

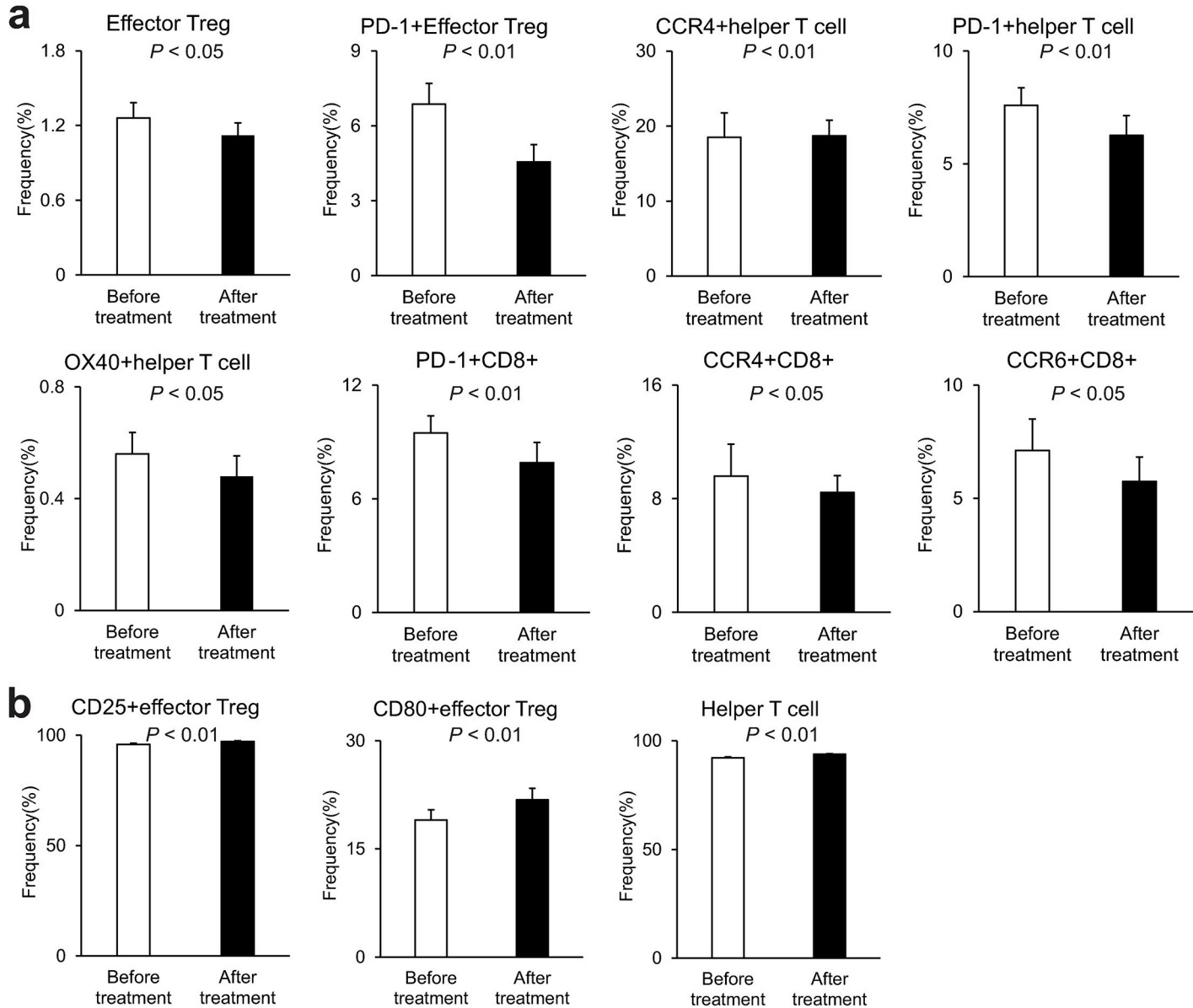

**Fig 3. Frequencies of peripheral blood immune cells with each profile in the SVR cases before and after OBV/PTV/r treatment.** Only significantly changed profiles are listed. (a) Significantly decreased profiles after treatment. (b) Significantly increased profiles after treatment. Scale bars are the median ± SE.

### T-box transcription factor 21 (TBX21) expression is involved in non-SVR status

RNA-seq analysis using PBMC RNA extracted from 31 patients showed that genes involved in the immune response were prominent in the SVR group before OBV/PTV/r treatment (Fig 5A), while genes involved in intracellular and extracellular signaling were prominent in PBMCs from the non-SVR group (Fig 5B). Moreover, we found that 67 genes were significantly expressed in the non-SVR cases compared with the SVR cases after OBV/PTV/r treatment (Fig 5C), and FACS analysis indicated that expression in PD-1+ Tregs in SVR cases was higher before OBV/PTV/r treatment. Therefore, we searched the National Center for Biotechnology Information gene database for genes related to PD-1+ Tregs, which were significantly

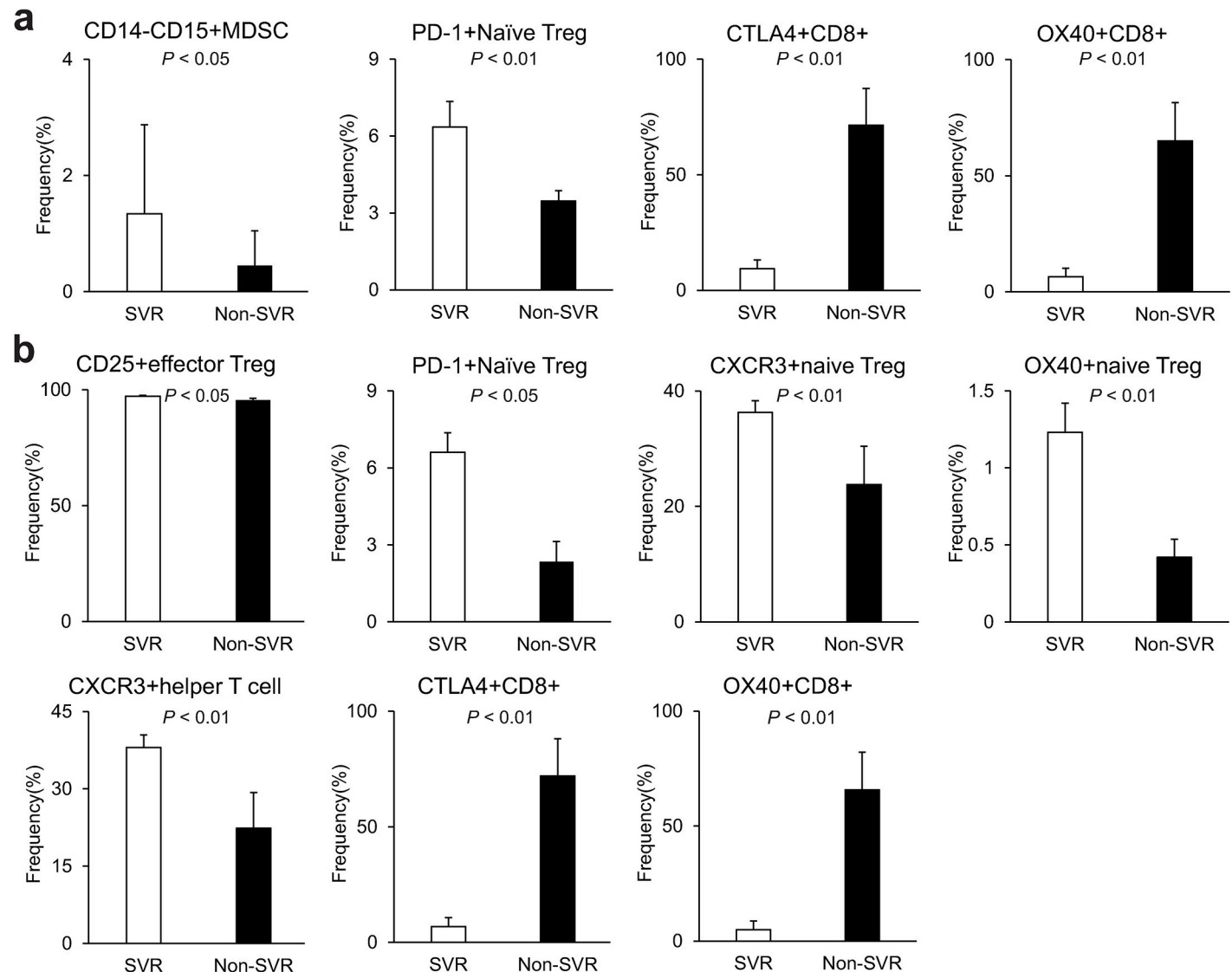

**Fig 4. Frequencies of peripheral blood immune cells with each profile in SVR and non-SVR cases.** Only significantly changed profiles are listed. (a) Before OBV/PTV/r treatment. (b) After OBV/PTV/r treatment. The data are the median ± SE.

different between the SVR and non-SVR groups before DAAs therapy, and PD-1+CD8+ T cells, which were significantly changed before and after DAAs therapy, and found 60 and 111 gene candidate, respectively (S9A and S9B Fig). When we searched for common genes among these genes, only TBX21 was found as a candidate (Figs 5C and S9). We confirmed TBX21 expression by qPCR; there was a significant correlation between the TBX21 expression level detected by RNA-seq and qPCR analyses (Fig 5D). Moreover, qPCR analysis indicated higher TBX21 expression in non-SVR cases than in SVR cases (Fig 5E). From these results, TBX21 was found to be involved with SVR and non-SVR status by OBV/PTV/r treatment.

## Discussion

HCV infection induces inefficient HCV-specific immune responses that lead to HCV persistence, which also exhibits hepatic carcinogenesis-related signaling. DAAs treatment for

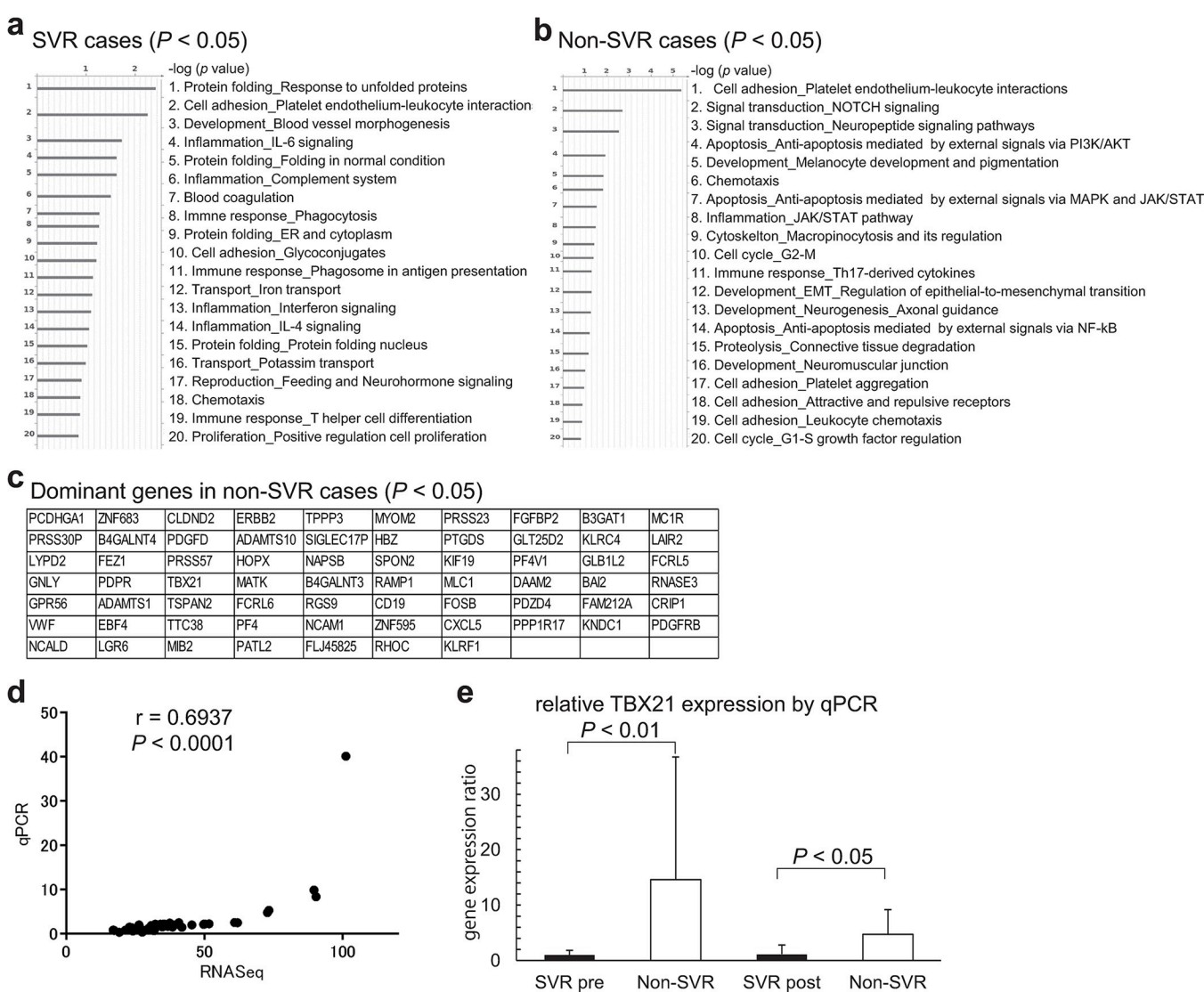

**Fig 5. Graphical representation of before OBV/PTV/r treatment by Gene Ontology (GO) enrichment of differentially expressed genes between the SVR and non-SVR groups and analysis of the candidate gene.** (a) Network objects that were arranged by the top 20 significant *P*-values in the SVR group. (b) Network objects that were arranged by the top 20 significant *P*-values in the non-SVR group. Bars represent the tendency of GO enrichment and number is the order of GO category. (c) Dominant genes in non-SVR cases before OBV/PTV/r treatment by RNA-seq analysis (*P* < 0.05). (d) Correlation of TBX21 gene expression between RNA-seq and qPCR analyses. (e) Relative TBX21 gene expression by qPCR analysis in SVR and non-SVR cases before and after treatment. Each non-SVR case is shown as the ratio of relative expression compared with SVR cases.

chronic HCV-infected patients has improved the rate of achieving an SVR to nearly 100% for all HCV genotypes. Moreover, there had been a problem with achieving an SVR according to mutations in the non-structural regions of the HCV genome, but recently introduced compounds have overcome this resistance problem. However, even in the SVR state, the risk of HCC remains, suggesting an association with residual inflammation even after HCV eradication. This situation can occur with any effective DAAs, which prompted us to analyze the specific immune cell profiles related to DAAs treatment.

Given that non-SVR cases have persistent HCV chronic inflammation after DAAs treatment, it is important to compare the immune profiles between non-SVR and SVR patients

who express distinct patterns after the successful eradication of HCV and with failure or untreated cases, and we can compare the immune profile changes associated with HCV infection, mainly using PBMCs. Identifying pre-treatment immune profiles associated with SVR or non-SVR could be critical for SVR prediction. SVR patients exhibit a decrease in PD-1 expression and effector Tregs as well as an increase in the CD4+ and CD8+ T cell profile. For non-SVR cases who fail to eradicate HCV, the opposite phenomenon might be observed. We observed a significant decrease in PD-1 + effector Tregs, but not in PD-1+ naïve Tregs, in SVR cases after DAAs treatment. Regarding the difference between naïve Tregs and effector Tregs, given the change from naïve to effector in response to HCV infection, we considered that the acquisition of specificity might be the cause of the observed PD-1 changes in effector Tregs. We thought that by comparing the immune profiles of peripheral blood and the gene expression profiles of PBMCs, we could identify novel factors. Our results represent a non-significant PD-L1-related immune profile before and after DAAs treatment in SVR cases such as those in which PD-L1 changes with HCV clearance, as recently reported [23]; however, there was a tendency toward decreased results (S3 Fig). It is possible that PD-L1 decreases in tissues as a change in immune profile during HCV clearance, although this cannot be detected by PBMC analysis.

Some DAAs exhibit a higher SVR rate in patients infected with any HCV genotype. Our experimental data indicate HCV-specific immune responses and can supply both HCV infection and eradication-related immune data that will generate more details about HCV-related inflammation signaling, which might be closely associated with hepatocarcinogenesis. To date, no associations between mutations in HCV non-structural regions and immune profiles have been observed. In this study, we also confirmed that single nucleotide polymorphisms in IL28B did not contribute to the SVR prediction-related immune profiles.

We compared the immune profiles with gene expression data generated using RNA-seq among PBMCs. There is a concern with this approach because RNA was isolated from all PBMCs for RNA-seq analysis, while the immune profiles were determined by FACS using the immune population of cells in PBMCs. An immune profile is not sufficiently relevant because it evaluates the proportion of the population of immune cells that exhibit a similar profile. However, we thought it would be possible to evaluate the gene clusters that had been identified in the immune profiles. According to the changes in these genes, we thought that it would be possible to compare the pathways that were significantly changed by comparing SVR and non-SVR cases. Our results suggested that TBX21 was a predictor of the non-SVR state. In fact, it has been pointed out that TBX21 is a T helper 1(Th1) cell-related transcription factor and is associated with IFN-related immune response [24–28]. However, the lymphocyte analyses in these cases, especially the RNAseq analysis, showed little difference in terms of IFN response. There was little correlation between TBX21 and IFN signaling-related genes in this study. Our findings showed that TBX21 was important for PD-1 related signaling involved in HCV F protein-mediated immune responses [29]. The results also showed that TBX21 was involved with PD-1, including in immune checkpoint signaling, but did not lead directly to IFN-related responses.

There are few reports of factors related to a non-SVR, except for HCV non-structural mutations, and since the SVR rate has been increased to nearly 100% with the current DAAs, the evaluation of an SVR is questionable. However, the immune regulatory genetic alterations involving a non-SVR obtained in this study are likely to provide significant factors for the identification of gene groups and pathways involved in developing chronic liver disease such as liver cirrhosis and related complications as well as the therapeutic effects of DAAs.

In this study, we analyzed a small number of non-SVR cases since the most recently developed DAAs have resulted in a high SVR rate, which makes it difficult to perform a comparison

between SVR and non-SVR cases. Therefore, it would be better to compare the populations that exist in both SVR and non-SVR cases when considering DAAs. Our experiments revealed a decrease in the immune profiles of PD-1+ effector Tregs, PD-1+ helper T cells, and PD-1+CD8+ T cells by successful HCV antiviral treatment. This makes sense of a study reporting an association between PD-1 expression and HCV-specific CD8+ T cell exhaustion in acute HCV infection [30]. In non-SVR cases, PD-1 expression in Tregs contributed to the persistence of HCV even with antiviral therapy, which may be a key modulator in addition to HCV resistance by antiviral therapy.

This is the reason why we considered non-SVR immune profiles or related gene candidates associated with the HCV infection-related immune response, and even if this is not correlated with obvious inflammation, these profiles may lead to unknown chronic inflammation or hepatocarcinogenesis. We identified TBX21 as a candidate gene, and it is related to HCV-encoded proteins, especially in the core region. Previous reports showed that TBX21 expression is highly correlated with HCV F protein [31–34], which is produced by a frameshift mutation in the HCV core region and produces a shortened polypeptide due to the generation of a stop codon. Other reports have shown associations between TBX21 with the immune response to viral infection or eradication or hepatocarcinogenesis. Thus, TBX21 is closely associated with SVR or non-SVR patterns induced by any antiviral drugs including DAAs or IFN. We analyzed this factor in detail with respect to the immune response that leads to hepatocarcinogenesis [29].

## Conclusions

In conclusion, we showed that the immune profiles of PD-1+ cells, CD4+ T cells, CD8+ T cells, and Tregs were more related before treatment in non-SVR cases treated with OBV/PTV/r, and that TBX21, but not HCV non-structural mutations, was associated with non-SVR status.

## Supporting information

**S1 Fig. Consolidated standard of reporting trials (CONSORT) diagram.** We enrolled 207 hepatitis C cases and analyzed 27 SVR and 4 non-SVR cases. OBV/PTV/r, ombitasvir/paritaprevir/ritonavir; SVR, sustained virological response.
(EPS)

**S2 Fig. Consolidated standard of reporting trials (CONSORT) diagram.** We enrolled 462 hepatitis C cases and analyzed 30 SVR and 10 non-SVR cases. DCV/ASV, daclatasvir and asunaprevir. SVR, sustained virological response; HCC, hepatocellular carcinoma.
(EPS)

**S3 Fig. Frequencies of peripheral blood immune cells with each profile in all patients before and after OBV/PTV/r therapy that did not change significantly.** The data are the median ± SE.
(EPS)

**S4 Fig. Frequencies of peripheral blood immune cells with each profile in all patients before and after OBV/PTV/r therapy by paired comparison analysis for each patient.** (a) decreased profiles after treatment. (b) increased profiles after treatment.
(EPS)

**S5 Fig. Frequencies of peripheral blood immune cells with each profile in SVR patients before and after OBV/PTV/r therapy by paired comparison analysis for each patient.** (a)

decreased profiles after treatment. (b) increased profiles after treatment.
(EPS)

**S6 Fig. Frequencies of peripheral blood immune cells with each profile in the non-SVR group before and after OBV/PTV/r therapy.** The data are the median ± SE.
(EPS)

**S7 Fig. Frequencies of peripheral blood immune cells with each profile in patients before and after DCV/ASV therapy.** (a) All patients. (b) SVR cases. (c) Non-SVR cases. The data are the median ± SE.
(EPS)

**S8 Fig. Frequencies of peripheral blood immune cells with each profile in SVR and non-SVR cases for DCV/ASV treatment.** (a) Before treatment of non-SVR cases with the Y93H/L31M mutation. (b) Before treatment of non-SVR cases without the Y93H/L31M mutation. (c) After treatment of non-SVR cases with the Y93H/L31M mutation. (d) After treatment of non-SVR cases without the Y93H/L31M mutation. The data are the median ± SE.
(EPS)

**S9 Fig. TBX21 determination from candidate genes by PD-1+Treg- or PD-1+CD8+-related immune profile.** (a) Candidate genes associated with PD-1 and Tregs based on a public database search. (b) Candidate genes associated with PD-1+CD8+ T cells based on a public database search.
(EPS)

**S1 Table. Patient characteristics for DCV/ASV treatment.**
(DOCX)

**S2 Table. Raw data of frequencies of peripheral blood immune cells with each profile for OBV/PTV/r treatment.**
(XLSX)

**S3 Table. Raw data of frequencies of peripheral blood immune cells with each profile for DCV/ASV treatment.**
(XLSX)

## Acknowledgments

We would like to thank N. Nishiyama, K. Fushimi, and M. Nakamura for technical assistance.

## Author Contributions

**Conceptualization:** Kazunori Kawaguchi, Shuichi Kaneko, Eishiro Mizukoshi, Taro Yamashita.

**Data curation:** Miyabi Miura, Michiko Nishino, Kazunori Kawaguchi, Shihui Li, Eishiro Mizukoshi.

**Formal analysis:** Miyabi Miura, Michiko Nishino, Kazunori Kawaguchi, Eishiro Mizukoshi.

**Funding acquisition:** Shuichi Kaneko, Eishiro Mizukoshi, Taro Yamashita.

**Investigation:** Tetsuro Shimakami, Toshikatsu Tamai, Hidetoshi Nakagawa, Takeshi Terashima, Noriho Iida, Hajime Takatori, Kuniaki Arai, Yoshio Sakai, Tatsuya Yamashita.

**Supervision:** Kazunori Kawaguchi, Masao Honda, Eishiro Mizukoshi, Taro Yamashita.

**Writing – original draft:** Miyabi Miura, Michiko Nishino.

**Writing – review & editing:** Kazunori Kawaguchi, Eishiro Mizukoshi.

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
