## [Decision Letter · Decision Letter 0]

18 Dec 2023

PONE-D-23-29904Programmed cell death-1 is involved with peripheral blood immune cell profiles in patients with hepatitis C virus antiviral therapyPLOS ONE

Dear Dr. Kawaguchi,

Thank you for submitting your manuscript to PLOS ONE. After careful consideration, we feel that it has merit but does not fully meet PLOS ONE’s publication criteria as it currently stands. Therefore, we invite you to submit a revised version of the manuscript that addresses the points raised during the review process.

We look forward to receiving your revised manuscript.

Kind regards,

Jason T. Blackard, PhD

Academic Editor

PLOS ONE

“The research leading to these results received funding from AbbVie Inc.”

4. We are unable to open your Supporting Information file [OBVPTVr_TS_09_SupplementaryFigure1.eps and OBVPTVr_TS_09_SupplementaryFigure6.eps (S1 Fig – S6 Fig)]. Please kindly revise as necessary and re-upload.

Additional Editor Comments:

This is an evaluation of immune profiles and factors predicting response / non-response to HCV direct acting agents in peripheral blood.

The intent of this study is quite interesting; however, the data presentation and discussion are a bit confusing and require some focus and significant clarification.

Table 1:  does “history of cancer” refer specifically to HCC or to all cancers?

Table 1:  “sex” should be changed to “gender”

Was HCV genotype determined before DAA initiation?  Reporting resistance mutations implies that these data are available but are not explicitly reported / stated.  Are immune cell profiles different by HCV genotype?  Or by gender or by age?

How many cells were used for RNAseq?

Are data normally distributed?  If not, non-parametric tests such as medians (rather than means) are most appropriate.  Given the small number of SVR cases included, the data are not likely to be normally distributed.

Reviewers' comments:

Reviewer's Responses to Questions

**Comments to the Author**

1. Is the manuscript technically sound, and do the data support the conclusions?

Reviewer #1: No

Reviewer #2: Partly

2. Has the statistical analysis been performed appropriately and rigorously? 

Reviewer #1: I Don't Know

Reviewer #2: Yes

3. Have the authors made all data underlying the findings in their manuscript fully available?

Reviewer #1: Yes

Reviewer #2: No

4. Is the manuscript presented in an intelligible fashion and written in standard English?

Reviewer #1: Yes

Reviewer #2: Yes

5. Review Comments to the Author

Reviewer #1: In this study, authors analyze peripheral blood immune cells in non-SVR compared to SVR, pre- and post-DAA. Understanding immune cell profiles pre-DAA and relating them to the outcome is important and could enable an understanding of why some perturbations persist in some individuals.

While the authors clearly introduce the problem, the discussion is unclear. While the focus is on non-SVR, the leading figures show SVR data. The main caveat is the minimal number of individuals in the non-SVR group, and hence, the lack of changes in non-SVR may be due to the lack of statistical power. Statistics are not well described (including for RNA-seq data) and need to be reviewed, as I am not sure using normal distribution tests is appropriate with such small numbers per group. Also, were comparisons paired (pre/post DAA per patient)?

Authors should comment on the lack of differences in PD-L1 expression. PD-1 ligands may be expressed differentially in response to treatment status on neutrophils that could not be studied here since PBMCs are used or on macrophages, which could not be assessed since they are located in tissues. In that matter, a recent report by Ang Cui et al. (J Hep 2023) examined liver myeloid cells in HCV pre- and post-DAA HCV cure and found overexpression of PD-L1 on neutrophils. This population contracted in SVR after DAA.

The discussion starts with the wrong statement: “HCV infection induces specific immune responses that lead to chronic infection, which also exhibits hepatic carcinogenesis-related signaling.”.…it’s the inefficient HCV-specific immune responses that lead to HCV persistence.

Furthermore, this first paragraph of the discussion ends with a sentence that does not provide a clear link to the next section nor to the study itself, as it goes as follows…”… even in the SVR state, the risk of HCC remains, suggesting an association with residual inflammation even after HCV eradication. This situation can occur with any effective DAAs, which prompted us to analyze the specific immune cell profiles related to DAA treatment.” And yet the next paragraph starts by “Therefore, it is important to compare the immune profiles between non-SVR and SVR patients…”.

Authors should be careful with overstatements and pay attention to clarity.

- For example, instead of …” It is more important to identify immune profile differences before DAA treatment between SVR and non-SVR patients that could lead to the prediction of successful HCV eradication by treatment.” Remove "more" or write something in line with... ” Identifying pre-treatment immune profiles associated with SVR or non-SVR could be critical for SVR prediction."

- On line 344. Instead of “The results indicated that TBX21 was a predictor of the non-SVR state,” say, “Our results suggest…”.

- Line 271…Authors observe higher PD-1 naïve Treg in SVR than in non-SVR pre-treatment and post-treatment. From there, they conclude that PD-1 expression in Tregs contributes to HCV eradication by antiviral therapy. How did authors come to this conclusion? In-depth discussion should be made about different patterns between naïve and effector Tregs, “the PD-1+ effector Treg being decreased after DAA in SVR only”

On line 283, instead of “our analysis showed significant differences in PD-1+ Tregs between the SVR and non-SVR,” remind the reader which differences were “higher or lower”

Also, to avoid confusion, use “higher or lower” when comparing groups at one state and "increased/decreased" when comparing pre/post.

Line 322-325. Some statements are confusing and need clarification. For example, the authors state, “It is more important to identify immune profile differences before DAA treatment between SVR and non-SVR patients that could lead to the prediction of successful HCV eradication by treatment. SVR patients exhibit a decrease in PD-1 expression and Tregs and an increase in the CD4+ and CD8+ T cell profile.” However, in the result section, they state that PD-1+ naïve Treg cells were higher in SVR than in non-SVR in pre-treatment, and PD-1+ naïve Treg and CD25+ effector Treg increased post-treatment in SVR.

Line 170, in the methods, it is helpful to readers to briefly mention what are the markers used…, e.g., chemokine receptor, immune checkpoints, etc.…

In conclusion, in this study, there needs to be more clarity in the result interpretation.

Reviewer #2: The study by Miura et al. explored the changes of immune cell subpopulations before and after DAA treatment in SVR and non-SVR patients. The authors found eTregs, PD-1+CD8+ T cells and PD-1+Helper T cells were decreased significantly in SVR after treatment, but no significant changes in non-SVR. Lastly, The authors found that TBX21 was associated with the non-SVR.The result is interesting. However, I have some concerns.

Major concerns

1. I am confused about how the cell populations and cell markers were selected for FACS sorting in this study. The order of cell sorting (maybe using arrows) should be shown in Schematic diagram (Fig1).

2. The authors found TBX21 were associated with the non-SVR. On the other hand,TBX21 is a Th1-related transcription factor and is associated with IFN-related immune response. How to explain the phenomena that TBX21 associated with both non-SVR and IFN-related immune response?

Minor concerns

1. Fig 1: The fraction of each cell population could be marked on figure.

2. The values of the y-axis are chaos. The fractions of cell subsets are not consistent and somtimes completely contradict each other for fig2, fig3, fig4.

E.g.

Fig2a, 2b:The y-axis (frequency) of the panel 1 and panel 3 of fig2b >100%)

Fig 4a: y-axis in the fist two panels of are low (1-9%), while y-axis in the last two panels are much high (70%), sum of which is >100%.

Fig 4b: y-axis in first panel >100%. The second panel and forth panel is much low.

3. Fig6: It looks weird to put the gene list in panel a,b,c

4. Fig6e the ylab is missing.

5. Abbreviations of DAA are incosistence: e.g. direct antiviral agent (DAA) in in Abstract; direct acting antivirals(DAAs) in Introduction.

6. The sequence data should be deposited into public database such as GEO

6. PLOS authors have the option to publish the peer review history of their article (what does this mean?). If published, this will include your full peer review and any attached files.

Reviewer #1: No

Reviewer #2: **Yes: **Wenfei Jin

---

## [Author Response · Author response to Decision Letter 0]

1 Feb 2024

<Response>

Thank you for this pointing out, we have carefully read the journal’s style requirements and revised the file names accordingly.

<Response>

　Thank you for pointing this out. We have now deposited our raw data in the GEO database.

https://www.ncbi.nlm.nih.gov/geo/query/acc.cgi?acc=GSE254592

(GEO reviewer token is “ytspccomppexxkn”.)

“The research leading to these results received funding from AbbVie Inc.”

<Response>

Thank you pointing this out. We have added the statement, “The funders had no role in study design, data collection and analysis, decision to publish, or preparation of the manuscript" to both the financial disclosure and the cover letter.

4. We are unable to open your Supporting Information file [OBVPTVr_TS_09_SupplementaryFigure1.eps and OBVPTVr_TS_09_SupplementaryFigure6.eps (S1 Fig – S6 Fig)]. Please kindly revise as necessary and re-upload.

<Response>

We apologize for the inconvenience. We have carefully saved these files as EPS format within 10MB file sizes that they can easily be opened.

Additional Editor Comments:

This is an evaluation of immune profiles and factors predicting response / non-response to HCV direct acting agents in peripheral blood.

The intent of this study is quite interesting; however, the data presentation and discussion are a bit confusing and require some focus and significant clarification.

Table 1: does “history of cancer” refer specifically to HCC or to all cancers?

<Response>

We apologize for the confusion. Here, “history of cancer” means that after HCC therapy and there was no active HCC at DAA treatment. We have revised Table 1 for clarity.

Table 1: “sex” should be changed to “gender”

<Response>

We have revised this accordingly.

Was HCV genotype determined before DAA initiation? Reporting resistance mutations implies that these data are available but are not explicitly reported / stated. Are immune cell profiles different by HCV genotype? Or by gender or by age?

<Response>

Thank you for pointing this out. We definitely checked the HCV genotype before DAA treatment. Among the cases that received OBV/PTV/r treatment, we did not observe resistance mutations such as Y93 and L31 in the NS5A position and have included this information in this manuscript. There were NS5A mutations among part of non-SVR cases, some profiles were similar regardless of the presence or absence of the Y93H mutation as shown in S8 Fig. Our results did not reveal any significant differences among HCV genotype, gender, and ages. 

How many cells were used for RNAseq?

<Response>

Thank you for pointing this out. We collected RNA from human whole blood, which ranged from 4.8 × 106 to1.1 × 107 cells. Because the amount was 2.5 ml per patient, we estimated that 1.2 × 107 to 2.8 × 107 cells per sample was collected during blood sampling. We have added text explaining this detail to the Methods section.

Are data normally distributed? If not, non-parametric tests such as medians (rather than means) are most appropriate. Given the small number of SVR cases included, the data are not likely to be normally distributed.

<Response>

As recommended, we have changed the values for the analyzed cases to medians; the results were unaffected by this change.

Reviewer #1: In this study, authors analyze peripheral blood immune cells in non-SVR compared to SVR, pre- and post-DAA. Understanding immune cell profiles pre-DAA and relating them to the outcome is important and could enable an understanding of why some perturbations persist in some individuals. 

While the authors clearly introduce the problem, the discussion is unclear. While the focus is on non-SVR, the leading figures show SVR data. The main caveat is the minimal number of individuals in the non-SVR group, and hence, the lack of changes in non-SVR may be due to the lack of statistical power. Statistics are not well described (including for RNA-seq data) and need to be reviewed, as I am not sure using normal distribution tests is appropriate with such small numbers per group. Also, were comparisons paired (pre/post DAA per patient)?

<Response>

Thank you for suggesting precise and essential points for improving the manuscript. It is true that the SVR rate with DAAs is high and there are very few non-SVR cases because the SVR rates are increasing, reaching almost all numbers. There are many DAAs for HCV antiviral treatment, and the SVR rate is increasing with each new agent, but the SVR rate is already high with initial DAAs such as DCV/ASV. The immune profile among peripheral blood lymphocytes following antiviral treatment is performed at 6 months after completion of DAA treatment. However, for non-SVR cases, FACS analysis was performed in this study when the patient was determined to be non-responsive, so it is difficult to compare SVR and non-SVR cases over the same time period. However, if there is a change in immune response, we thought we should consider changes in immune profiles before and after treatment by analyzing various subsets. The paired comparisons suggested by the reviewer were similarly changed; we have included these details in Supplemental Figures 4 and 5 as well as in the Results section. The number of non-SVR cases looks too small compared with the number of SVR cases, and there appears to be cause for concern about the statistical power based on the differences in each group. However, in general, the more unbalanced the number of cases, the less statistically significant the difference, suggesting that the differences that do appear have a much greater influence on significance. As pointed out, despite the statistical imprecision, it is suggested that any significant differences detected can be concluded to be more strongly different.

Authors should comment on the lack of differences in PD-L1 expression. PD-1 ligands may be expressed differentially in response to treatment status on neutrophils that could not be studied here since PBMCs are used or on macrophages, which could not be assessed since they are located in tissues. In that matter, a recent report by Ang Cui et al. (J Hep 2023) examined liver myeloid cells in HCV pre- and post-DAA HCV cure and found overexpression of PD-L1 on neutrophils. This population contracted in SVR after DAA.

<Response>

Thank you for pointing out the lack of PD-L1 changes with HCV clearance in our study and introducing a related report. As the reviewer mentioned, there was no significant change after DAAs treatment in the profiles of CD14+PD-L1 and CD14-CD15+PD-L1, as shown in Supplementary Figure 3. We consider it possible that PD-L1 decreases in tissues as a change in immune profile during HCV SVR, as with DAAs treatment because we analyzed peripheral blood lymphocytes and there was a weakly significant difference compared with the liver tissue analysis. We have cited the recommended article in the Discussion section and commented about PD-L1 changes.

The discussion starts with the wrong statement: “HCV infection induces specific immune responses that lead to chronic infection, which also exhibits hepatic carcinogenesis-related signaling.”.…it’s the inefficient HCV-specific immune responses that lead to HCV persistence.

<Response>

Thank you for pointing this out. We have revised the sentence as “insufficient HCV-specific immune responses that lead to HCV persistence”.

Furthermore, this first paragraph of the discussion ends with a sentence that does not provide a clear link to the next section nor to the study itself, as it goes as follows…”… even in the SVR state, the risk of HCC remains, suggesting an association with residual inflammation even after HCV eradication. This situation can occur with any effective DAAs, which prompted us to analyze the specific immune cell profiles related to DAA treatment.” And yet the next paragraph starts by “Therefore, it is important to compare the immune profiles between non-SVR and SVR patients…”.

<Response>

As the reviewer pointed out, there is insufficient explanation in the discussion section, so we have added the sentence: “Given that non-SVR cases have persistent HCV chronic inflammation after DAAs treatment, ...”

Authors should be careful with overstatements and pay attention to clarity.

- For example, instead of …” It is more important to identify immune profile differences before DAA treatment between SVR and non-SVR patients that could lead to the prediction of successful HCV eradication by treatment.” Remove "more" or write something in line with... ” Identifying pre-treatment immune profiles associated with SVR or non-SVR could be critical for SVR prediction."

<Response>

Thank you for pointing this out. We have revised this as “Identifying pre-treatment immune profiles associated with SVR or non-SVR could be critical for SVR prediction.” 

- On line 344. Instead of “The results indicated that TBX21 was a predictor of the non-SVR state,” say, “Our results suggest…”.

<Response>

In accordance with the reviewer’s instruction, we have changed this to “Our results suggested…”.

- Line 271…Authors observe higher PD-1 naïve Treg in SVR than in non-SVR pre-treatment and post-treatment. From there, they conclude that PD-1 expression in Tregs contributes to HCV eradication by antiviral therapy. How did authors come to this conclusion? In-depth discussion should be made about different patterns between naïve and effector Tregs, “the PD-1+ effector Treg being decreased after DAA in SVR only”

<Response>

The OBV/PTV/r results in this report show that PD-1+naïve Tregs in SVR and non-SVR, both before (Figure 4a) and after (Figure 4b) DAA treatment, have a relatively higher rate of SVR compared with non-SVR. However, DCV/ASV exhibited a similar trend. PD-1+effector Tregs did not significantly differ between SVR and non-SVR before or after DAA treatment, even if they were relatively higher before in the DAA cases in the SVR group compared with the non-SVR group. Regarding the difference between naïve Tregs and effector Tregs, given the change from naive to effector in response to HCV infection, we considered that the acquisition of specificity might be the cause of the observed PD-1 changes in effector Tregs. We have added text explaining this matter to the Discussion section.

On line 283, instead of “our analysis showed significant differences in PD-1+ Tregs between the SVR and non-SVR,” remind the reader which differences were “higher or lower”

Also, to avoid confusion, use “higher or lower” when comparing groups at one state and "increased/decreased" when comparing pre/post.

<Response>

Thank you for pointing this out. We have revised the sentence as “FACS analysis indicated that expression in PD-1+ Tregs in SVR cases was higher before OBV/PTV/r treatment”.

Line 322-325. Some statements are confusing and need clarification. For example, the authors state, “It is more important to identify immune profile differences before DAA treatment between SVR and non-SVR patients that could lead to the prediction of successful HCV eradication by treatment. SVR patients exhibit a decrease in PD-1 expression and Tregs and an increase in the CD4+ and CD8+ T cell profile.” However, in the result section, they state that PD-1+ naïve Treg cells were higher in SVR than in non-SVR in pre-treatment, and PD-1+ naïve Treg and CD25+ effector Treg increased post-treatment in SVR.

<Response>

Based on the reviewer’s comment, we have changed this to “SVR patients exhibit a decrease in PD-1 expression and effector Tregs” for clarification. As the reviewer pointed out, the explanation “PD-1+ naïve Treg cells were higher in SVR than in non-SVR in pre-treatment, and PD-1+ naïve Treg and CD25+ effector Treg increased post-treatment in SVR” was confusing, so we have modified the explanation of the results in Figure 4b by including “compared with the non-SVR group”.

Line 170, in the methods, it is helpful to readers to briefly mention what are the markers used…, e.g., chemokine receptor, immune checkpoints, etc.…

<Response>

Thank you for this important suggestion. We have added an explanation of antibodies in FACS analysis to the Methods section.

In conclusion, in this study, there needs to be more clarity in the result interpretation.

<Response>

Thank you for pointing this out. There was a lack of uniformity in the interpretation of the results, and your suggestion has helped to clarify this. We have checked the explanation again, including for any inconsistencies, and revised it based on the reviewers’ comments.

Reviewer #2: The study by Miura et al. explored the changes of immune cell subpopulations before and after DAA treatment in SVR and non-SVR patients. The authors found eTregs, PD-1+CD8+ T cells and PD-1+Helper T cells were decreased significantly in SVR after treatment, but no significant changes in non-SVR. Lastly, The authors found that TBX21 was associated with the non-SVR.The result is interesting. However, I have some concerns.

Major concerns

1. I am confused about how the cell populations and cell markers were selected for FACS sorting in this study. The order of cell sorting (maybe using arrows) should be shown in Schematic diagram (Fig1).

<Response>

Thank you for pointing this out. Because it is important to include more information on FACS sorting in Fig 1, we have revised the figure as a schematic diagram and updated the figure legend accordingly.

2. The authors found TBX21 were associated with the non-SVR. On the other hand,TBX21 is a Th1-related transcription factor and is associated with IFN-related immune response. How to explain the phenomena that TBX21 associated with both non-SVR and IFN-related immune response?

<Response>

Thank you for pointing this out. We showed that TBX21 gene was upregulated in non-SVR cases by RNA-seq analysis. As the reviewer pointed out, TBX21 is a Th1-related transcription factor and is also involved in IFN-related immune response such as IFN-gamma. However, the lymphocyte analyses in these cases, especially the RNA-seq analysis, showed little difference in terms of IFN response. There was little correlation between TBX21 and IFN signaling-related genes in this study. Our findings showed that TBX21 was important for PD-1 related signaling involved in HCV F protein-mediated immune responses. The results also showed that TBX21 was involved with PD-1, including in immune checkpoint signaling, but did not lead directly to IFN-related responses. These very important results have been reflected in the manuscript, particularly in the Discussion section.

Minor concerns

1. Fig 1: The fraction of each cell population could be marked on figure.

<Response>

Thank you for pointing this out. We have added the fraction of each cell p

---

## [Decision Letter · Decision Letter 1]

12 Feb 2024

Programmed cell death-1 is involved with peripheral blood immune cell profiles in patients with hepatitis C virus antiviral therapy

PONE-D-23-29904R1

Dear Dr. Kawaguchi,

We’re pleased to inform you that your manuscript has been judged scientifically suitable for publication and will be formally accepted for publication once it meets all outstanding technical requirements.

Kind regards,

Yury E Khudyakov, PhD

Academic Editor

PLOS ONE

Additional Editor Comments (optional):

Reviewers' comments:

Reviewer's Responses to Questions

**Comments to the Author**

1. If the authors have adequately addressed your comments raised in a previous round of review and you feel that this manuscript is now acceptable for publication, you may indicate that here to bypass the “Comments to the Author” section, enter your conflict of interest statement in the “Confidential to Editor” section, and submit your "Accept" recommendation.

Reviewer #2: All comments have been addressed

2. Is the manuscript technically sound, and do the data support the conclusions?

Reviewer #2: Yes

3. Has the statistical analysis been performed appropriately and rigorously? 

Reviewer #2: N/A

4. Have the authors made all data underlying the findings in their manuscript fully available?

Reviewer #2: Yes

5. Is the manuscript presented in an intelligible fashion and written in standard English?

Reviewer #2: Yes

6. Review Comments to the Author

Reviewer #2: Most of my concerns have been addressed in some way. I do suggest this manuscript could be published.

7. PLOS authors have the option to publish the peer review history of their article (what does this mean?). If published, this will include your full peer review and any attached files.

Reviewer #2: **Yes: **Wenfei Jin

---

## [Editor Report · Acceptance letter]

14 May 2024

PONE-D-23-29904R1 

PLOS ONE

Dear Dr. Kawaguchi, 

I'm pleased to inform you that your manuscript has been deemed suitable for publication in PLOS ONE. Congratulations! Your manuscript is now being handed over to our production team.

Kind regards, 

on behalf of

Dr. Yury E Khudyakov 

Academic Editor

PLOS ONE